# Ruminal Yeast Strain with Probiotic Potential: Isolation and Characterization and Its Effect on Rumen Fermentation In Vitro

**DOI:** 10.3390/microorganisms13061270

**Published:** 2025-05-30

**Authors:** Pin Song, Xiaoran Yang, Manman Hou, Yue Chen, Liping Liu, Yuyan Feng, Yingdong Ni

**Affiliations:** Key Laboratory of Animal Physiology & Biochemistry, Nanjing Agricultural University, Nanjing 210095, China; 2020207051@stu.njau.edu.cn (P.S.); 2022107032@stu.njau.edu.cn (X.Y.); 2021207007@stu.njau.edu.cn (M.H.); 2024207005@stu.njau.edu.cn (Y.C.); 2022107033@stu.njau.edu.cn (L.L.); fyy@stu.njau.edu.cn (Y.F.)

**Keywords:** *Candida rugosa*, ruminal fermentation, probiotic yeast, microbial community composition

## Abstract

The objective of this study is to isolate, identify, and describe rumen yeast strains and assess their probiotic potentials and effects on ruminal fermentation in vitro. Yeasts were isolated from ruminal fluids, yielding 59 strains from nine distinct species. A number of tests were conducted to assess their anaerobic traits, growth rate, acid tolerance, and lactate utilization ability, and a second screening in fresh ruminal fluid to evaluate in vitro pH and acid accumulation was conducted. The probiotic yeast *Candida rugosa* (NJ-5) was selected for in vitro culture studies on rumen fermentation. Finally, *Candida rugosa* (NJ-5) with good probiotic characteristics was chosen to investigate its effects on ruminal fermentation in vitro. The batch culture technique was used to explore the effects of *Candida rugosa* (NJ-5) yeast culture on rumen fermentation parameters. By altering the fermentation substrate to a concentrate-to-roughage ratio of 70:30, which simulated a high-concentration diet. The CON, LYC, MYC, and HYC groups were supplemented with 0%, 1%, 2%, and 5% *Candida rugosa* (NJ-5) yeast culture (dry matter basis), respectively. The pH value and volatile fatty acid (VFA) contents were determined at 6, 12, and 24 h after fermentation. The results showed that adding *Candida rugosa* (NJ-5) yeast culture successfully modulated in vitro rumen fermentation. Compared to the CON group, HYC had a significantly mitigated reduction in pH in fermentation, resulting in a significant increase in total VFAs and acetate levels (*p* < 0.05). Additionally, 16S rRNA sequencing revealed that *Candida rugosa* (NJ-5) yeast culture supplementation did not significantly alter ruminal bacterial alpha diversity (*p* > 0.05). At the phylum and genus taxonomic levels, *Candida rugosa* (NJ-5) yeast culture addition increased the relative abundance of several functionally important bacterial groups in the rumen microbial community. Compared to the CON group, the HYC group concurrently had an increased abundance of Desulfobacterota, *Christensenellaceae_R-7_group*, *F082*, and *Ruminococcus* (*p* < 0.05) but a significantly reduced abundance of Cyanobacteria, Bdellovibrionota, *Succinivibrionaceae_UCG-002*, *Enterobacter*, and *Succinivibrio* (*p* < 0.05). The in vitro fermentation experiment demonstrated that the optimal dry matter supplementation of *Candida rugosa* (NJ-5) into the basal diet was 5%, which could be effective for maintaining ruminal fermentation stability when ruminants were fed a high-concentrate diet. This study provides empirical support for the use of yeast as a nutritional supplement in ruminant livestock management, as well as a theoretical underpinning for further animal research.

## 1. Introduction

Recent advancements in ruminant production have prominently incorporated yeast products as microbial feed supplements [1]. Numerous studies suggest that adding yeast culture substances into ruminant diets may increase feed intake, improve rumen fermentation, stabilize ruminal pH, and strengthen the immune system of animals, hence promoting production performance [2,3,4]. Moreover, yeast can compete with infections through competitive inhibition pathogen growth in the digestive tract [5,6]. Currently, the majority of commercial yeast products utilized in cattle production are derived from *Saccharomyces cerevisiae* strains [7]. Nevertheless, these strains were not explicitly selected for their probiotic characteristics in ruminants. It has been reported that *Saccharomyces cerevisiae* strains show limited growth in the rumen and cannot reproduce stably in this environment [8]. Therefore, when selecting potential probiotics, it is crucial to isolate strains from the same ecological niche in which they will be carefully selected to be suitable for the specific host to ensure reliable probiotic functions [9,10]. Key criteria for selecting a yeast strain include its ability to thrive and demonstrate vigorous growth in the rumen environment with the capacity to tolerate ruminal pH, enhance rumen fermentation, and ultimately improve rumen functionality while maintaining the stability of the microbial community. The in vitro ruminal fermentation method is commonly utilized to assess the nutritional effectiveness of feed additives on rumen fermentation before in vivo studies [11,12]. Previous studies suggest that *Pichia kudriavzevii*, *Candida rugosa*, and *Kodamaea ohmeri*, derived from the ruminal fluid of dairy cows, could serve as effective probiotic supplements by regulating lactic acid metabolism in dairy cattle nutrition [13]. Additionally, *Candida tropicalis* and *Candida norvegensis*, isolated from the rumen, significantly improved ruminal fermentation [14]. It has been reported that *Candida norvegensis* derived from the rumen significantly improved the in vitro fermentation properties of oat straw, promoting microbial growth and the rate of dry matter decomposition, but greatly reduced methane emissions [15]. Similarly, *Pichia kudriavzevii*, isolated from ruminal fluid, can influence specific microbial communities to enhance fiber and fat digestion and also promote acetate-type fermentation processes [16]. Prior research indicated that supplementation with active dry yeast failed to restore the original microbiota composition that existed before the commencement of SARA in dairy calves, implying the restricted functional effectiveness of this commercial yeast strain [17]. This evidence underscores that the efficacy of probiotics is contingent upon their adaptation to certain ecological niches. *Candida* has been recognized as a natural species in bovine gastrointestinal tracts, and its probiotic potential has been experimentally confirmed [18,19]. As of now, the sole study by Sirisan et al. has shown the application of *Candida rugosa* in rumen fermentation research [13]. Nevertheless, no subsequent research has produced culture products from *Candida rugosa* or conducted in vitro dose–response trials. In this study, the method in vitro fermentation was employed to evaluate the effects of yeast strains isolated and selected from the goat rumen on ruminal fermentation, and 16S rRNA sequencing was used to investigate the influence of the most promising probiotic strains on the bacterial community. The aim of this study was to explore the potential genetic resources from the rumen and provide basic data for the application of these probiotics for improving the welfare of animals and productivity in ruminants and non-ruminants.

## 2. Materials and Methods

The animal experiments were approved by the Institutional Animal Care and Use Committee of Nanjing Agricultural University according to the Guidelines on Ethical Treatment of Experimental Animals (2006) No. 398 set by the Ministry of Science and Technology (Beijing, China, 2006) (IACUC Approval No. NJAU.20221101207).

### 2.1. Collection of Rumen Samples for Yeast Isolation and Identification

Fresh ruminal fluid samples were collected to isolate yeasts from the ventral sac of the rumen via the rumen fistula before the morning feeding. The rumen fluid was filtered through four layers of gauze, stored in a CO_2_-filled vacuum flask to maintain an anaerobic environment, and swiftly brought to the laboratory for yeast isolation. The filtered rumen fluid was serially diluted 10 times from 10^−1^ to 10^−5^ with 0.85% sterile saline. Following dilution, the fluid was inoculated into YPD agar plates and incubated at 39 °C for 48 h. The YPD agar (1000 mL) consisted of yeast extract 5 g, peptone 10 g, glucose 20 g, and agar 14 g. Morphological observations were performed, and colonies displaying diverse shapes and colors were selected using an inoculation loop. The streak plate method for three rounds of purification was used until the recovered yeast strains exhibited consistent morphological characteristics.

After 48 h of incubation, a total of 59 yeast isolates were cultured in vitro, and their cellular biomass was collected. Genomic DNA from yeast was extracted using a Yeast Genome DNA Extraction Kit (Beijing Solarbio Science & Technology Co., Ltd., Beijing, China). A PCR was performed with primers ITS1(5′-TCCGTAGGTGAACCTGCGG-3) and ITS4 (5′-TCCTCCGCTTATTGATATGC-3′) [20]. The PCR system was as follows: 25 μL of 2 × *Taq* Master Mix (Nanjing Novozan Biotechnology Co., Ltd., Nanjing, China), 1 μL template DNA, 1 μL each of forward and reverse primers (10 μM), and ddH_2_O to a final volume of 50 μL. The PCR amplification program was as follows: initial denaturation at 95 °C for 5 min; 30 cycles of 95 °C for 30 s, annealing at 55 °C for 30 s, and extension at 72 °C for 30 s; and final extension at 72 °C for 10 min. The PCR amplified products were subjected to electrophoresis on a 1% agarose gel. The amplicons were sent to Nanjing Tsingke Biotech Co., Ltd. (Nanjing, China) for DNA sequencing. The sequences were aligned using BLAST (https://blast.ncbi.nlm.nih.gov/Blast.cgi, retrieved 1 January 2024) on the NCBI GenBank platform. The sequences were acquired and analyzed to construct a phylogenetic tree using MEGA software (version 7.0).

### 2.2. Selection of Yeasts with Probiotic Potential

The viability in anaerobic circumstances at 39 °C, lactic acid assimilation capacity, and acid tolerance ability were used to assess the 59 isolated yeast strains for probiotic suitability in ruminants. After each test, only the yeasts that showed the best results were put through the next test.

Initially, the proliferation of the 59 isolated yeast strains under anaerobic circumstances was evaluated. Following three generations of activation, the yeast strains were injected at a 1% (*v*/*v*) concentration into YPD liquid medium and cultivated at 39 °C, 150 rpm for 48 h. The yeast suspension was subsequently inoculated onto YPD agar plates with an inoculation loop, after which the plates were sealed in anaerobic bags and incubated at 39 °C for 48 h. The expansion of colonies was documented and recorded. 

To identify yeast strains exhibiting accelerated growth rates, we chose those possessing anaerobic capabilities and inoculated them into YPD liquid medium at a 1% (*v*/*v*) concentration. Following incubation at 39 °C and 150 rpm for 48 h, we collected the fermentation liquid and assessed the absorbance at 600 nm with a microplate reader. Yeast strains with an OD600 value exceeding 1.0 were chosen for additional investigations. Subsequently, yeast strains with high acid tolerance were selected and then inoculated onto YPD liquid media under pH 5.0, 5.4, and 5.8, utilizing a 1% (*v*/*v*) inoculum. Following incubation at 39 °C with shaking at 150 rpm for 48 h, The supernatant was sampled and was quantified at absorbance at 600 nm using a microplate reader. Yeast strains with an OD600 value exceeding 0.8 were selected and preserved for subsequent investigations. 

Finally, lactic acid was included to modify the YPD liquid medium to a concentration of 15 mmol/L. The chosen yeast strains were inoculated into the medium at a 1% (*v*/*v*) concentration and cultivated at 39 °C with agitation at 150 rpm for 48 h. The supernatant was sampled, and the level of lactic acid was quantified using a commercial assay kit (Nanjing Jiancheng Bioengineering Institute, Nanjing, China). The lactic acid consumption rate was determined using the following formula: (initial lactic acid concentration in the medium—lactic acid concentration in the fermentation liquid)/initial lactic acid concentration in the medium.

### 2.3. The Preparation of Candida Rugosa (NJ-5) Yeast Culture

*Candida rugosa* (NJ-5) was inoculated into the YPD liquid medium, then incubated at 39 °C with shaking at 150 rpm for 12 h to produce *Candida rugosa* (NJ-5) seed culture for subsequent application. A 5% inoculum of the yeast seed culture was subsequently introduced to the YPD liquid medium for aerobic fermentation, and the mixture was incubated at 39 °C with shaking at 150 rpm for 24 h. Subsequently, the aerobic fermentation liquid was introduced to a sterile grain solid medium at a 10% inoculation rate. The solid medium consisted of wheat bran (80%), soybean meal (13%), and corn flour (7%). The mixture was enveloped with three layers of sterile gauze and a lid and then incubated at 30 °C for 72 h. The yeast culture was uniformly distributed on the porcelain tray and air-dried at ambient temperature. The desiccated material was pulverized and sifted through a 40–60 mesh to obtain the yeast culture.

### 2.4. Experimental Design and In Vitro Culture Procedure

Three healthy goats (average body weight 31 ± 1.3 kg) fitted with permanent rumen cannulas were used as ruminal fluid donors. The goats were fed twice a day, at 8:00 AM and 6:00 PM, with free access to food and water throughout the day. The composition and nutritional components of the daily diet fed to the goats are shown in Table 1. Before morning feeding, ruminal fluids were collected and stored in well-insulated and securely sealed thermos bottles preheated to 39 °C. Carbon dioxide gas was put into the bottles, and air was evacuated prior to sealing the lid. Ruminal fluids were promptly filtered through four layers of gauze. The fermentation substrates utilized in this experiment included the following: 0.49 g of maize, 0.21 g of soybean meal, 0.15 g of oats, and 0.15 g of alfalfa, amounting to a total of 1 g (the composition and nutritional levels of the fermentation substrate are shown in Appendix A). The formula of this fermentation substrate simulates the proportion of the high-concentrate diet feeding practice in the intensive system. In vitro rumen fermentation trial 1: The yeast strain group (NJ-5, NJ-12, NJ-14, NJ-36, NJ-46) was supplemented with 1 mL of prepared yeast suspension, achieving a yeast inoculation level of 2 × 10^6^ CFU/mL, with three biological replicates per group. In vitro rumen fermentation trial 2: yeast culture concentrations of 0%, 1%, 2%, and 5% based on dry matter were incorporated into the fermentation substrate, establishing the control group (CON), low-dose group (LYC), middle-dose group (MYC), and high-dose group (HYC), with three replicates for each group. Artificial saliva was prepared as reported previously by Menke et al. [12]. After thorough mixing, the amalgamation was put in a 39 °C constant temperature water bath with a continuous supply of CO_2_. The combination was subsequently blended in a 1:2 ratio of rumen fluid to artificial saliva, and 60 mL of the mixture was transferred to 100 mL fermentation bottles, with continuous CO_2_ flow to sustain anaerobic conditions, and was cultured at 39 °C with shaking at 150 rpm. The supernatant was sampled at incubated for 6, 12, and 24 h for the assessment of the pH value and fermentation parameters. For VFA measurement, samples were centrifuged at 8000× *g* for 15 min at 4 °C, then filtered through a 0.45 µm membrane to collect the supernatant.

### 2.5. Assessment of VFAs

The VFAs were measured using a GC-14B (Shimadzu, Shijota, Japan) to assess the amounts of acetic acid, propionic acid, butyric acid, isovaleric acid, and valeric acid as previously conducted [21]. The concentration of lactic acid was quantified by a commercial test kit (A109-21, Nanjing Jiancheng Bioengineering Institute) by measuring the absorbance at 530 nm.

### 2.6. DNA Extraction and 16S rDNA Sequencing

The CTAB approach, as outlined by Denman et al. [22], was employed to extract total DNA from rumen fluid samples. Following quality control, the PCR amplification of the bacterial 16S rRNA gene V4 region was conducted with universal bacterial primers 515F: 5′-GTGCCAGCMGCCGCGG-3′ and 806R: 5′-GGACTACHVGGGTWTCTAAT-3′. The PCR products underwent purification, followed by library building and sequencing using the Novaseq 6000 platform (Meiji Biomedical Technology Co., Ltd., Shanghai, China). Following rigorous filtering and database comparison, high-quality data were acquired for further research. The sequences were categorized into operational taxonomic units (OTUs) with Uparse software (v7.0.1001). The operational taxonomic units (OTUs) were annotated utilizing the Mothur methodology and compared against the SILVA138 SSU rRNA database to acquire taxonomic data and assess microbial abundance across the hierarchical levels of kingdom, phylum, class, order, family, genus, and species. An α-diversity analysis (including the Chao1 index, Shannon index, Simpson index, quantity of phyla, and quantity of genera) and principal coordinate analysis (PCoA) were conducted. Tukey’s test and the Wilcoxon test were used to evaluate intergroup differences.

### 2.7. Data Analysis

Before analysis, all data were tested for normality and homogeneity of variance. All the data were processed using Excel 2016 and analyzed using IBM SPSS Statistics 26 (IBM, Armonk, NY, USA). A one-way analysis of variance (ANOVA) was performed for comparisons of more than two groups, and Duncan’s multiple comparison was used to compare the differences between groups. The data are displayed as means and standard errors (SEM), and *p* < 0.05 indicates statistical significance.

## 3. Results

### 3.1. Isolation and Identification of Yeast Strains

A total of 59 yeast strains (NJ 1~59) derived from goat ruminal fluid were isolated under aerobic circumstances, as indicated in Table 2, and then the ITS region of the isolated yeast strains was amplified via PCR and subsequently sequenced. A phylogenetic tree was generated by sequencing based on Genbank data (Figure 1). These 59 isolated yeast strains were classified into nine distinct species: *Candida rugosa* (32.2%), *Pichia kudriavzevii* (20.3%), *Trichosporon asahii* (15.3%), *Candida tropicalis* (10.2%), *Magnusiomyces capitatus* (6.8%), *Candida pararugosa* (6.8%), *Meyerozyma caribbica* (5.1%), *Sporidiobolus pararoseus* (1.7%), and *Yarrowia lipolytica* (1.7%).

### 3.2. Anaerobic Capacity of Yeast Strains

The results showed that 37 out of 59 yeast strains could proliferate in an anaerobic environment, which were chosen for further screening experiments. Figure 2 shows the growth capacity of the selected 37 yeast strains. The OD600 values varied from 0.4 to 1.7 (Appendix A), with 18 strains exceeding an OD600 value of 1, representing 48.6% of the total strains, which demonstrated a robust growth potential and chosen for subsequent testing.

### 3.3. Evaluation of Acid Tolerance of Yeast Strains

The selected 18 yeast strains were cultured under the conditions of pH 5.0, 5.4, and 5.8 to evaluate acid tolerance. As shown in Figure 3, all these yeast strains could endure low-pH conditions (pH < 5.8), and 10 yeast strains exhibited OD600 values greater than 0.8 (Appendix A), which were finally chosen for further assessment.

### 3.4. Evaluation of Lactate Utilization Capacity of Yeast Strains

In order to measure the lactate utilization capability, the selected 10 yeast strains were cultured in lactate-enriched media. Figure 4 showed that all yeast strains demonstrated varying capacities for lactate consumption, with assimilation rates between 16% and 60%. Among these strains, five strains exhibited lactate utilization rates exceeding 50% and then were chosen for the subsequent tests.

### 3.5. Effect of Yeast Strains on VFA Production in In Vitro Rumen Fermentation

As shown in Table 3, compared to the control group (CON), only the NJ-14 yeast strain showed a significant increase in total volatile fatty acid (TVFA) concentration (*p* < 0.05). The strains of NJ-5 and NJ-12 had markedly elevated acetate concentrations (*p* < 0.05), and all five strains had a greatly enhanced production of propionate (*p* < 0.05) but significantly reduced valerate and isobutyrate levels (*p* < 0.05). Moreover, the yeast strains of NJ-14, NJ-36, and NJ-46 had markedly decreased ratios of acetate to propionate (*p* < 0.05). After 24 h of fermentation, the pH value produced by the NJ-5 yeast strain was markedly elevated compared to the CON group (*p* < 0.05); thus, NJ-5 was chosen for further investigation.

### 3.6. Preparation of Yeast Culture

In this study, the prepared yeast fermentation product exhibited a brown, powdery consistency; consistent coloration; the absence of unpleasant odors; and no apparent mold or contaminants (Figure 5).

### 3.7. Impact of Yeast Culture Supplementation on pH Value in In Vitro Fermentation

As shown in Table 4, compared to the CON group, the MYC group and HYC group had significantly increased pH values in the fermented liquids after 6 h culture, and after 24 h culture, yeast treatment groups all showed higher levels of pH compared to CON (*p* < 0.05).

### 3.8. Effects of Yeast Culture on VFA Concentrations in In Vitro Fermentation

As shown in Table 5, compared to the CON group, the total concentration of VFAs in the HYC group was markedly elevated, but the proportion of butyrate was dramatically reduced (*p* < 0.05). The LYC, MYC, and HYC groups exhibited considerably elevated percentages of acetate, and the ratio of acetate to propionate was higher (*p* < 0.05); however, the proportion of propionate was markedly reduced compared to CON.

### 3.9. Impact of Yeast Culture on Bacterial Community in In Vitro Fermentation

#### 3.9.1. Rumen Bacterial Diversity Analysis

The in vitro rumen fermentation liquid microbiota was characterized and quantitatively analyzed via 16S rDNA sequencing. A Venn diagram showed that there were 3070 ASVs in the CON group, 3100 in the LYC group, 3778 in the MYC group, and 3305 in the HYC group, with 489 ASVs common to all four groups (Figure 6A). The alpha diversity index revealed that the Chao1, Shannon, Pielou_e and Simpson indices exhibited no statistically significant differences that were noted between the groups (*p* > 0.05) (Figure 6B). The bacterial communities of the LYC, MYC, and HYC groups exhibited distinct clustering (Figure 6C), signifying analogous microbial compositions within these groups, and adding yeast culture enriched the bacterial community in the in vitro fermentation medium.

#### 3.9.2. Impact of Yeast Culture on Bacteria Abundance in In Vitro Fermentation

As shown in Figure 7, the predominant phyla in the in vitro rumen fermentation fluids were Firmicutes, Bacteroidetes, and Proteobacteria (Figure 7A). Figure 7C,D use heatmaps to show the different abundances of bacterial communities at the phylum and genus levels in different treatment groups. In comparison to the CON group, the LYC, MYC, and HYC groups exhibited a significant increase in the relative abundance of Desulfobacterota (*p* < 0.05) but a significant decrease in the relative abundance of Cyanobacteria (*p* < 0.05) (Figure 7A,E). The LYC group exhibited a substantial increase in the relative abundance of Fibrobacterota (*p* < 0.05), whereas the MYC and HYC groups had a significant decrease in the relative abundance of Bdellovibrionota (*p* < 0.05) (Figure 7A,E). At the genus level, the LYC and MYC groups had a considerably augmented relative abundance of *Ruminococcus, Selenomonas*, and *Rumiobacter* compared to the CON group (*p* < 0.05) and a dramatically reduced relative abundance of *Succinivibrioaceae_UCG-002* and *Succinivibrio* (*p* < 0.05) (Figure 7B,F). The HYC group exhibited a considerable increase in the relative abundance of *Christensenell-aceae_R-7_group*, *F082*, and *Ruminococcus* (*p* < 0.05) and a considerably decreased relative abundance of *Succinivibrioaceae_UCG-002*, *Enterobacter*, and *Succinivibrio* compared to CON (*p* < 0.05) (Figure 7B,F).

The linear discriminant analysis effect size (LEfSe) analysis (*p* < 0.05, LDA > 2.0) revealed a significant difference in the relative abundance of the 15 detected bacterial taxa (Figure 8). The order Bacillales, order Bifidobacteriales, family Bifidobacteriaceae, family Enterobacteriaceae, genus *Bifidobacterium*, and genus *Enterobacter* were the dominant bacterial groups enriched in the CON group. The order Oscillospirales, order Peptostreptococcales_Tissierellales, family Anaerovoracaceae, genus *Family_XIII_AD3011_group*, genus *NK4A214_group*, and genus *Prevotellaceae_NK3B31_group* were enriched in the LYC group. The family Bacteroidales_RF16_group and genus *Bacteroidales_RF16_group* were enriched in the HYC group.

## 4. Discussion

In this study, a total of 59 yeast strains were successfully isolated from goat ruminal fluids and classified into nine yeast species. Previous studies showed that yeasts from the genus Candida (*Candida albicans, Candida parapsilosis, Candida tropicalis*, and *Candida rugosa*) [23], *Trichosporon asahii* [24], and *Pichia kudriavzevii* [25,26] can be isolated from the rumen contents of ruminants. Consistently, in the present study, among the isolated species, *Candida parapsilosis*, *Candida rugosa*, and *Picha kudriavzevii* have been previously recognized for their probiotic potential. *Saccharomyces cerevisiae* is commonly used as a probiotic in ruminant nutrition [27], which was unfortunately not isolated in this study.

Previous studies have isolated *Candida* from the bovine gastrointestinal tract and confirmed its potential as a feed additive for cattle through specific physiological and cytological screening [18,19]. However, these studies have not validated the practical application effects of *Candida rugosa* in animal production through in vitro rumen fermentation experiments or in vivo animal trials. In ruminant production, yeast is typically applied in the form of yeast culture [28]. In this study, the probiotic candidate strain *Candida rugosa* (NJ-5) obtained through isolation and screening was prepared into yeast culture, and its effects on the fermentation characteristics of high-concentrate substrates were systematically evaluated using in vitro rumen fermentation experiments. The results demonstrated that *Candida rugosa* (NJ-5) yeast culture effectively alleviated the pH decline in fermentation broth induced by high-concentrate substrates and improved rumen fermentation, suggesting its potential application in mitigating SARA induced by high-concentrate diets in animal production.

Under optimum conditions, the rumen maintains an anaerobic environment, sustaining a temperature of 38.5 to 40.0 °C and a pH of 6.0 to 7.0. Subacute ruminal acidosis (SARA) arises when the rumen pH declines below 5.8, leading to alterations in the rumen microbial composition. The aim of this study was to develop yeast strains capable of thriving in anaerobic rumen environments, demonstrating vigorous growth at 39 °C, adapting to the low-pH conditions typical of SARA, and efficiently metabolizing lactate. A total of 10 out of 59 isolated yeast strains demonstrated consistent survival under SARA conditions, as determined by the evaluations of anaerobic tolerance, growth potential, and low pH adaptation. The collection included five strains of *Candida rugosa*, three strains of *Picha kudriavzevii*, and one strain each of *Candida pararugosa, Candida tropicalis*, and *Trichosporon asahii*. Previous studies demonstrate that *Candida rugosa, Picha kudriavzevii*, and *Candida pararugosa* exhibit anaerobic tolerance and acid resistance, which was consistent with our findings [24].The rumen-derived yeast strain *Candida rugosa* (NJ-5) exhibits excellent biological characteristics, including robust growth activity, strong anaerobic tolerance, notable acid resistance, and efficient lactate assimilation. In an in vitro rumen fermentation trial, live-cell suspensions of *Candida rugosa* (NJ-5) were cultured, and the results demonstrated that after 24 h of fermentation, the pH of the NJ-5 treatment group was significantly higher than that of the control, indicating its ability to effectively mitigate ruminal pH decline. Additionally, this strain significantly improved rumen fermentation patterns by increasing the proportion of acetate in the fermentation broth. Based on these findings, NJ-5 was selected as the optimal candidate strain due to its remarkable advantages in modulating rumen fermentation parameters.

It is well known that under SARA conditions, lactic acid accumulates in the rumen, leading to a reduction in pH and negatively impacting rumen health. Therefore, reducing lactic acid levels is crucial for maintaining rumen pH balance, enhancing rumen fermentation, and increasing ruminant productivity. Our findings revealed that *Candida rugosa, Picha kudriavzevii, Candida tropicalis*, and *Trichosporon asahii* had varying capacities for lactate utilization, with *Candida rugosa, Picha kudriavzevii*, and *Candida pararugosa* achieving utilization rates exceeding 50%. Similarly, significant populations of *Picha kudriavzevii* were detectable in the rumen of dairy cows suffering from SARA, showcasing its strong lactate-utilizing ability [29]. The condition of the rumen is crucial for the general well-being and production of ruminants. The pH of the rumen is a vital indicator of fermentation status, with the optimal range being 6.2~6.8 [30,31]. Numerous studies suggest that yeast supplementation may reduce lactate accumulation and enhance microbial diversity, hence increasing rumen pH [32,33]. In our experiment, the addition of five different yeast strains did not significantly alter the pH of in vitro rumen fermentation fluid, as shown in previous reports [34,35]. It is reasonable to speculate that dietary supplementation with yeast products derived from ruminal fluids can protect ruminants against SARA suffering due to the higher utilization of lactic acid.

Among the selected strains, *Candida rugosa* (NJ-5) was identified as the most promising probiotic candidate. This strain has been previously isolated from the rumen of dairy cattle and has exhibited probiotic activity [13,19]. In agreement with the beneficial traits discovered in our study, this is the first study to report that *Candida rugosa* (NJ-5) shows potential as a probiotic for alleviating SARA caused by high-concentrate diets. Yeast culture products are generated by first cultivating the yeast strain in a liquid medium, then performing solid-state fermentation, and then drying the outcome. These products consist of living yeast cells, extracellular metabolites (such as vitamins and enzymes), and the denatured fermentation media [33]. Yeast culture is widely employed as a sustainable feed additive in ruminant production owing to its positive effects on productive performance [36,37]. Currently, most commercial yeast culture products employ *Saccharomyces cerevisiae*, with improved strains being selected or genetically engineered. Yeasts originating from the rumen are more capable of flourishing in a ruminal low-pH and anaerobic environment and more effective at managing rumen microbial populations. In this study, the yeast strain of *Candida rugosa* was chosen for its ability to alleviate SARA caused by high-concentrate diets. Under normal conditions, the pH of the rumen is maintained at roughly 6.5. However, when ruminants are fed with a high-concentrate diet, this easily leads to the accumulation of both VFAs and lactic acid, thus lowering the pH value below 5.8 and finally causing SARA [38]. Our experimental results demonstrated that the addition of yeast culture led to a dose-dependent increase in the pH of fermentation fluid. Adding 5% yeast culture can successfully mitigate the pH decrease, which was consistent with the results of a study conducted on sheep [39].

It has been reported that yeast products can enhance the activity of rumen bacteria in metabolizing lactic acid into VFAs, thereby reducing lactate accumulation and elevating the ruminal pH value [40]. Volatile fatty acids (VFAs) produced by rumen bacteria serve as a crucial energy source for ruminants. Adding yeast to the diet may change the types of bacteria in the rumen, helping the bacteria that break down fiber, which in turn changes the kinds and amounts of certain volatile fatty acids (VFAs) made in the rumen [41]. Studies demonstrate that yeast supplementation in dairy cows can significantly increase the concentrations of acetate, propionate, and total VFAs [42]. Propionate, as an essential gluconeogenic precursor, significantly influences energy metabolism in ruminants [43]. The modification of rumen fermentation processes induced by high-grain diets is a key contributor to SARA [44]. Research demonstrates that high-grain feeding modifies rumen fermentation from an acetate-dominant pattern to one characterized by propionate and butyrate dominance [45]. This study revealed that yeast culture significantly increased total VFA and acetate levels while markedly reducing valerate levels; however, no significant changes were seen in the concentrations of valerate, isovalerate, or isobutyrate. The data suggest that yeast culture modifies the fermentation pattern of high-concentrate substrates from propionate dominance to acetate dominance, thereby partially mitigating the effects of high-concentrate diets.

Han et al. [43] found that the addition of yeast culture in sheep improved the diversity and richness of rumen microbiota. Similarly, it has been reported that adding yeast fermentation products to dairy cows reduced the microbial diversity caused by SARA [46]. However, our results showed that yeast culture did not have a big effect on the variety and abundance of microbiota in the in vitro fermentation system, as reported by Dai et al. [47]. The different effects of yeast culture on the α-diversity of rumen microbes could be due to different animal models, food types, yeast cultures, and dosages. The primary bacterial phyla in this experiment were Bacteroidota, Firmicutes, and Proteobacteria. The phylum Proteobacteria includes some pathogenic species, such as Escherichia and Salmonella, associated with gastrointestinal diseases and inflammation. The HYC group showed a reduction in Proteobacteria abundance compared to the CON group, suggesting that high-dose yeast cultivation could potentially enhance animal immunity [48]. Fibrobacterota is an essential fibrolytic phylum in the rumen, adept at converting low-quality feed into volatile fatty acids [49]. Desulfobacterota are sulfate-reducing bacteria (SRB) that can convert sulfate into hydrogen sulfide [50,51]. In comparison to the control group, the addition of *Candida rugosa* (NJ-5) yeast culture markedly enhanced the relative abundance of Desulfobacterota, potentially indicating a self-regulatory response within the rumen microenvironment. Organisms within the phylum Cyanobacteria are sporadically identified in ruminants; however, they generally represent a negligible proportion (less than 1%) of the microbial community. These bacteria are frequently located in soil and aquatic ecosystems, and their functional significance in the rumen is still ambiguous [52]. Their presence may be associated with restricted oxygen transport into the rumen or the fermentation of polysaccharides under microaerobic conditions [53]. *Christensenellaceae_R-7* is chiefly linked to the management of the gut microbiota and contributes to host immune regulation and metabolic homeostasis [54]. This taxon has been demonstrated to promote rumen growth and boost nutrient digestion and absorption in ruminants [55]. Dietary supplementation with *Candida rugosa* (NJ-5) yeast culture markedly enhanced the relative abundance of *Christensenellaceae_R-7* in comparison to the control, indicating the selective encouragement of advantageous microbial communities.

At the genus level, the abundance of *Streptococcus* was significantly reduced by three doses of YC supplementation. Meanwhile, *Selenomonas*, a well-known genus that breaks down lactate, showed a noticeable rise in the LYC and MYC groups and a higher relative abundance in the HYC group compared to the control group. These results indicate that YC may alleviate SARA disease by promoting the growth of bacteria that use lactate as a substrate, such as *Selenomonas*, and then decreasing the amount of lactate in the rumen. Additionally, YC significantly decreased the abundance of *Succinivibrionaceae_UCG-001*, which primarily produces succinate and competes with methanogens for hydrogen [56]. The relative abundance of *F082* and *Fibrobacter* was also increased in the YC-supplemented groups compared to the control, indicating that YC may boost fiber degradation by fostering the growth and activity of fibrolytic bacteria, which ultimately improve animal welfare and productive performance.

In vitro rumen fermentation is a widely used, simplified technique that effectively mimics real rumen fermentation, offering advantages such as operational simplicity, standardization ease, and high reproducibility. However, this method has inherent limitations; it cannot fully replicate the complex rumen environment within a single vessel, such as the absence of absorption dynamics and the rumen–intestine passage and the differences between free-living and epimural bacterial populations. Since the experimental findings are derived from in vitro conditions, they may not fully reflect in vivo rumen fermentation in live animals. Thus, to further evaluate the potential applications of *Candida rugosa* (NJ-5), additional animal studies are warranted.

## 5. Conclusions

In this study, a total of 59 yeast strains were successfully isolated from goat ruminal fluids, and *Candida rugosa* (NJ-5) was selected as the most promising probiotic yeast to improve the fermented environment in the rumen particularly under high-concentrate diets. In fact, adding *Candida rugosa* (NJ-5) yeast culture can mitigate the reduction in pH in fermentation, increasing the total VFA and acetate levels through remodeling the rumen microbial composition with a higher abundance of Desulfobacterota, *Christensenellaceae_R-7_group*, *F082*, and *Ruminococcus* but a lower level of Cyanobacteria, Bdellovibrionota, *Succinivibrioaceae_UCG-002*, *Enterobacter*, and *Succinivibrio*. Therefore, rumen native yeast Candida rugosa (NJ-5) has significant potential in mitigating SARA induced by high-concentrate diets, providing both an excellent microbial strain and a theoretical basis for developing probiotics to prevent and treat SARA in ruminants.

## Figures and Tables

**Figure 1 microorganisms-13-01270-f001:**
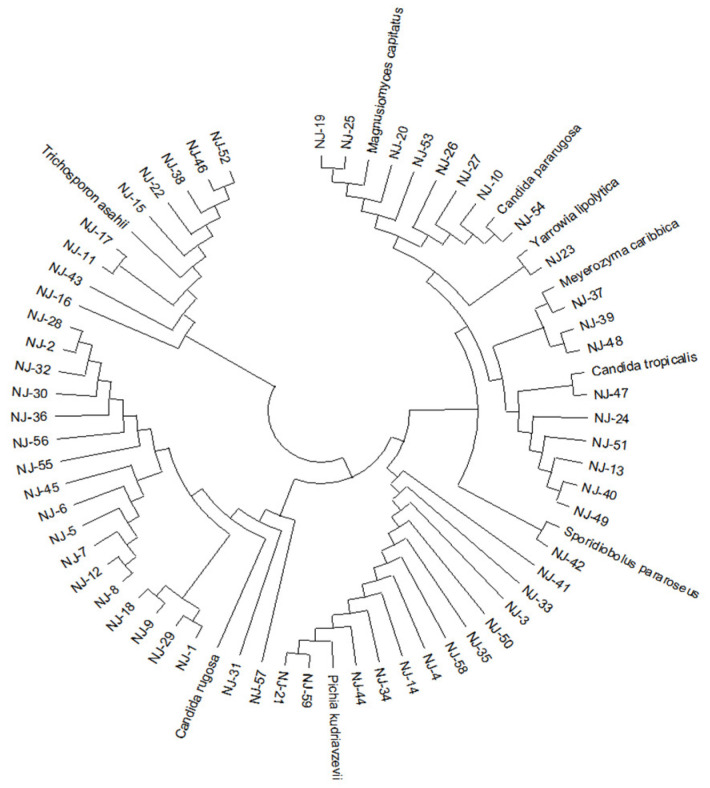
Phylogenetic tree construction of yeast strains.

**Figure 2 microorganisms-13-01270-f002:**
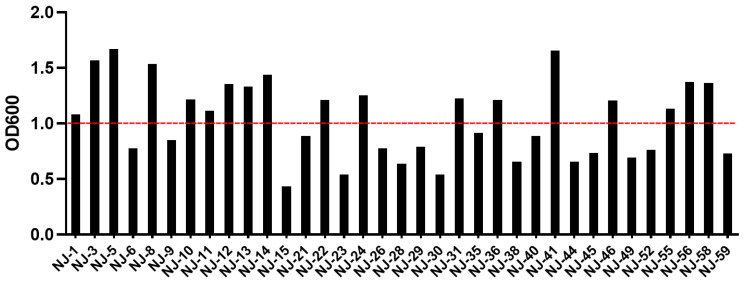
Growth capacity of yeast strains.

**Figure 3 microorganisms-13-01270-f003:**
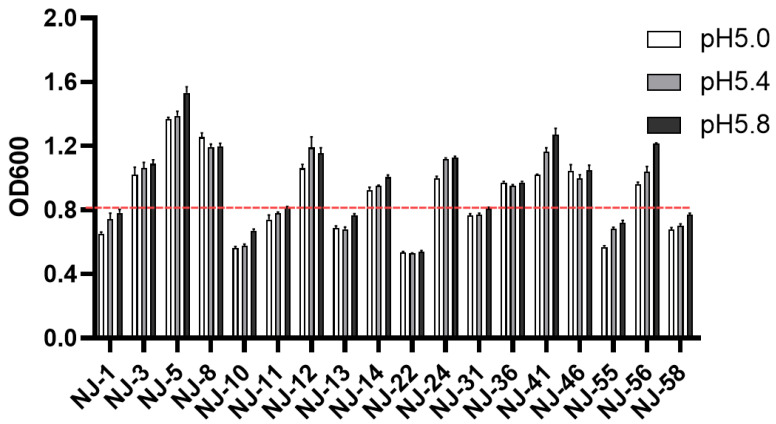
Acid resistance capacity of yeast strains.

**Figure 4 microorganisms-13-01270-f004:**
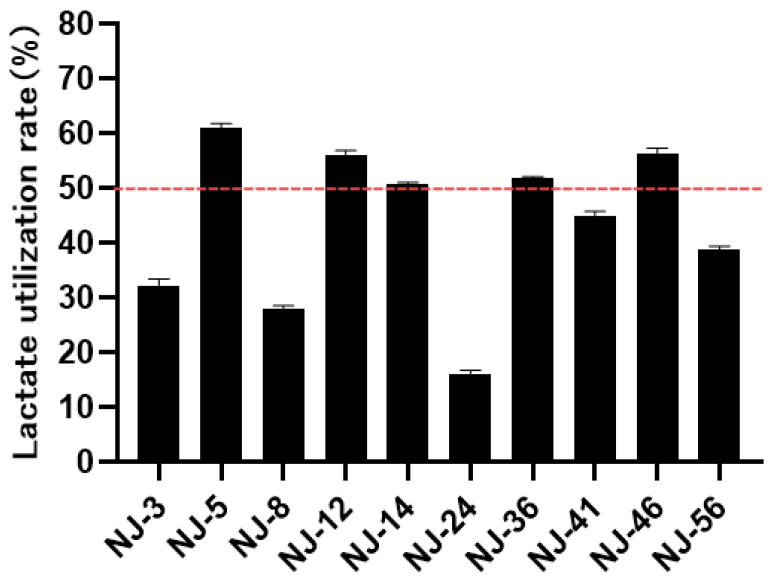
Lactate utilization capacity of yeast strains.

**Figure 5 microorganisms-13-01270-f005:**
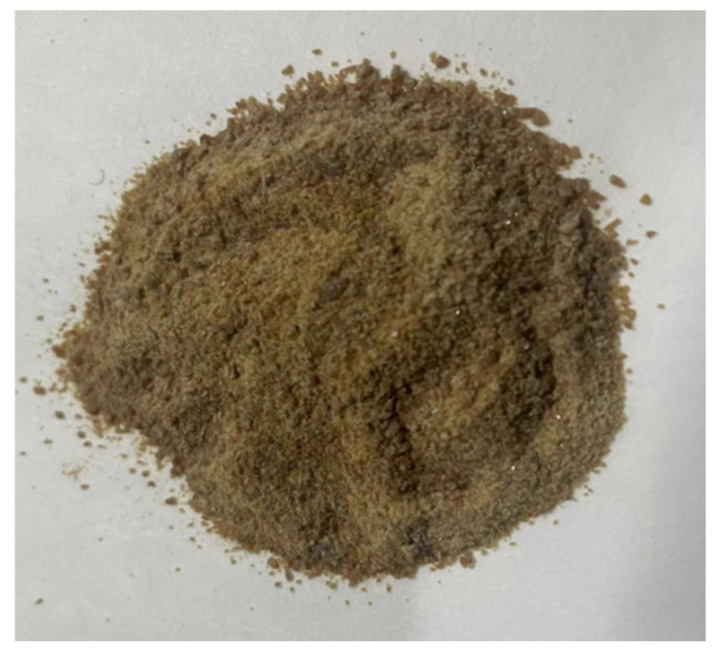
Picture of yeast culture sample.

**Figure 6 microorganisms-13-01270-f006:**
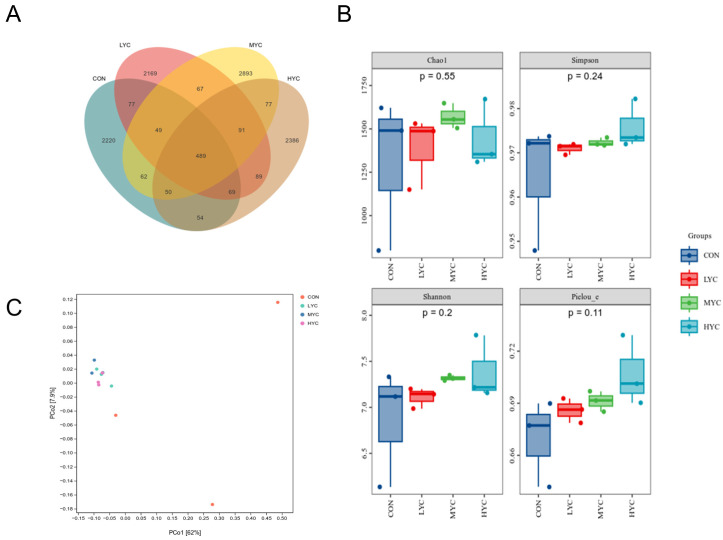
Rumen bacterial diversity analysis. (**A**) Venn diagram showing number of ASVs. (**B**) Alpha diversity including Chao1, Shannon, Pielou_e, and Simpson diversity indices for ASVs. (**C**) Beta diversity analysis based on PLS-DA analysis of ASVs.

**Figure 7 microorganisms-13-01270-f007:**
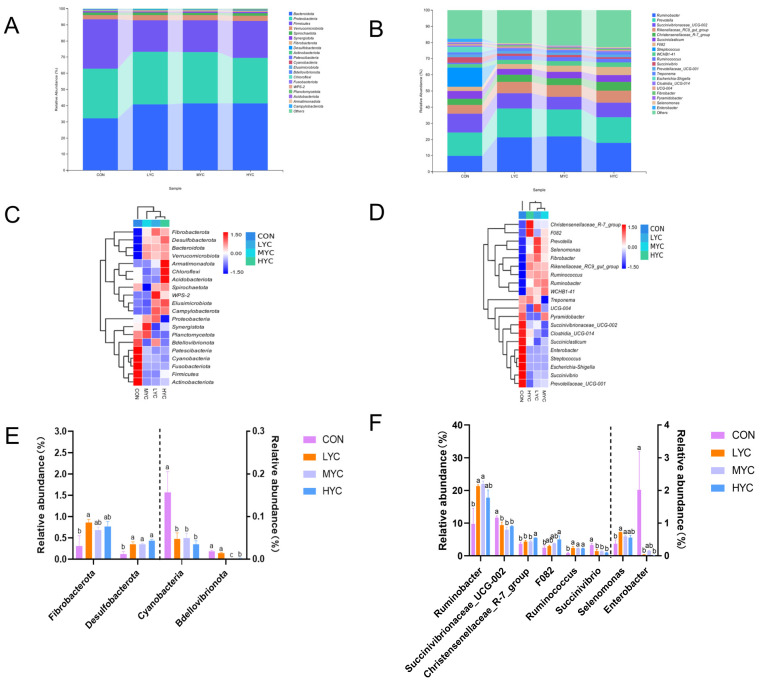
Effects of adding yeast culture on bacterial community in goat in vitro rumen fermentation fluid. Relative abundance of bacteria at phylum level (**A**) and genus level (**B**). Heatmap of species composition at phylum level (**C**) and genus level (**D**). Differential abundance at phylum level (**E**) and genus level (**F**).^a,b,c^ Means in row with different superscript letters differ significantly at *p* < 0.05.

**Figure 8 microorganisms-13-01270-f008:**
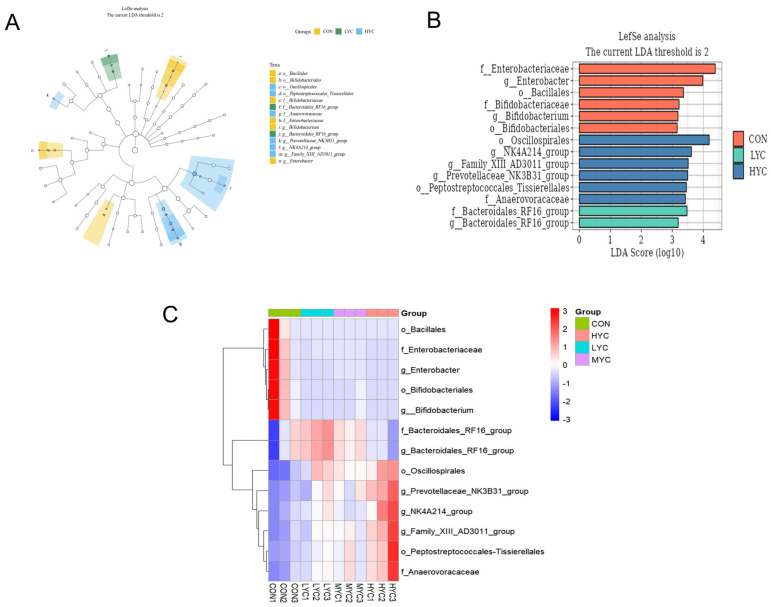
Differences in the composition of the ruminal microbiome among different groups. The differential abundant bacteria among the groups are presented in the LDA cladogram generated via LEfSe analysis (*p* < 0.05, LDA > 2.0) (**A**,**B**). A heatmap showing significantly different microbiota at the order, family, and genus levels (**C**).

**Table 1 microorganisms-13-01270-t001:** Composition and nutrient levels of basal diet (DM basis).

Diet Ingredients %		Nutritional Composition	
Alfalfa	70	ME ^2^ (MJ/kg)	8.81
Corn	14	CP (%)	10.81
Soybean meal	13	NDF (%)	44.82
Calcium hydrophosphate	1.42	ADF (%)	23.89
Limestone	0.58	Starch (%)	23.16
NaCl	0.5	Calcium (%)	0.81
Premix ^1^	0.5	Phosphorus (%)	0.47
Total	100		

^1^ The premix provided the following per kg of diets: MnSO_4_ 153 mg, ZnSO_4_ 186 mg, FeSO_4_ 125 mg, CoCl_2_ 8.25 mg, KIO_3_ 25 mg, CuSO_4_ 33 mg, NaSeO_3_ 4 mg, VA 15.28 mg, VE 0.47 mg. ^2^ ME was a calculated value, while the others were measured values.

**Table 2 microorganisms-13-01270-t002:** The 59 yeast strains investigated in the present study.

Yeast Species	Number of Isolates	Strain ID
*Candida rugosa*	19	NJ (1, 2, 5–9, 12, 18, 28–32, 36, 45, 55–57)
*Pichia kudriavzevii*	12	NJ (3, 4, 14, 21, 33–35, 41, 44, 50, 58, 59)
*Trichosporon asahii*	9	NJ (11, 15–17, 22, 38, 43, 46, 52)
*Candida tropicalis*	6	NJ (13, 24, 40, 47, 49, 51)
*Magnusiomyces capitatus*	4	NJ (19, 20, 25, 53)
*Candida pararugosa*	4	NJ (10, 26, 27, 54)
*Meyerozyma caribbica*	3	NJ (37, 39, 48)
*Sporidiobolus pararoseus*	1	NJ (42)
*Yarrowia lipolytica*	1	NJ (23)

**Table 3 microorganisms-13-01270-t003:** Effects of 5 different strains of yeast on rumen fermentation parameters.

Culture Time (h)	Groups						SEM	*p*-Values
	CON	NJ-5	NJ-12	NJ-14	NJ-36	NJ-46		
pH	5.58 ^b^	5.65 ^a^	5.61 ^ab^	5.57 ^b^	5.61 ^ab^	5.58 ^b^	0.05	0.040
Total VFA, mmol/L	100.13 ^bc^	101.02 ^ab^	100.82 ^abc^	101.41 ^a^	100.64 ^abc^	100.06 ^c^	0.28	0.003
Acetate (%)	52.61 ^c^	53.34 ^a^	53.23 ^ab^	52.72 ^bc^	52.73 ^bc^	52.84 ^abc^	0.16	0.003
Propionate (%)	27.7 ^c^	28.37 ^b^	28.39 ^ab^	28.81 ^a^	28.45 ^ab^	28.69 ^ab^	0.13	0.000
Butynate (%)	13.87	13.52	13.56	13.47	13.66	13.40	0.17	0.155
Isobutyrate (%)	1.62 ^a^	1.40 ^b^	1.38 ^b^	1.42 ^b^	1.45 ^b^	1.45 ^b^	0.05	0.002
Valerate (%)	2.63 ^a^	1.98 ^c^	1.97 ^c^	2.08 ^bc^	2.19 ^b^	2.04 ^bc^	0.06	0.000
Isovalerate (%)	1.56	1.51	1.51	1.50	1.52	1.50	0.04	0.654
Acetate/Propionate ratio	1.90 ^a^	1.88 ^ab^	1.88 ^ab^	1.83 ^c^	1.85 ^bc^	1.84 ^bc^	0.01	0.000

SEM, standard error of mean. ^a,b,c^ Means in row with different superscript letters differ significantly at *p* < 0.05.

**Table 4 microorganisms-13-01270-t004:** Effect of yeast culture on in vitro fermentation fluid pH value for goats.

Culture Time (h)	Groups				SEM	*p*-Values
	CON	LYC	MYC	HYC		
6	5.89 ^b^	5.92 ^ab^	6.07 ^a^	6.03 ^a^	0.06	0.038
12	5.85 ^b^	5.84 ^b^	5.85 ^b^	6.00 ^a^	0.03	0.001
24	5.69 ^b^	5.84 ^a^	5.82 ^a^	5.88 ^a^	0.05	0.040

SEM, standard error of mean. ^a,b^ Means in row with different superscript letters differ significantly at *p* < 0.05.

**Table 5 microorganisms-13-01270-t005:** Effects of yeast culture on fermentation parameters in rumen fermentation broth of goat in vitro.

Culture Time (h)	Groups				SEM	*p*-Values
	CON	LYC	MYC	HYC		
Total VFA, mmol/L	75.69 ^b^	76.00 ^ab^	76.09 ^ab^	76.36 ^a^	0.16	0.018
Acetate (%)	53.53 ^c^	54.56 ^b^	55.09 ^b^	56.01 ^a^	0.16	0.000
Propionate (%)	28.62 ^a^	27.41 ^b^	27.19 ^b^	26.70 ^c^	0.10	0.000
Butynate (%)	14.46 ^a^	14.62 ^a^	14.43 ^a^	14.09 ^b^	0.10	0.005
Isobutyrate (%)	1.12	1.08	1.09	1.04	0.04	0.214
Valerate (%)	1.20	1.29	1.21	1.12	0.05	0.090
Isovalerate (%)	1.05	1.03	0.99	1.03	0.04	0.543
Acetate/Propionate ratio	1.97	1.99	1.92	1.91	0.01	0.001

SEM, standard error of mean. ^a,b,c^ Means in row with different superscript letters differ significantly at *p* < 0.05.

## Data Availability

The data presented in this study are available on request from the corresponding author. The data are not publicly available due to privacy.

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
