# Peer review of "Ruminal Yeast Strain with Probiotic Potential: Isolation and Characterization and Its Effect on Rumen Fermentation In Vitro"

_microorganisms, 2025, doi:10.3390/microorganisms13061270_

Round 1

Reviewer 1 Report

Comments and Suggestions for Authors

The present study, entitled "Ruminal Yeast Strain with Probiotic Potential: Isolation and Characterization and Its Effect on Rumen Fermentation In Vitro," holds particular relevance in the field of animal science research, as it contributes to the identification and characterization of indigenous yeast strains with probiotic potential specifically adapted to the ruminal environment. The use of Candida rugosa (NJ-5) highlights an innovative approach to positively modulate ruminal fermentation, particularly under high-concentrate diet conditions, offering new perspectives for improving ruminant health and production efficiency. Furthermore, this work provides a solid experimental basis for future in vivo studies. However, several revisions could enhance the quality of the manuscript.

Author Response

Comments 1: The idea of isolating ruminal yeast strains with probiotic potential is timely and relevant, especially for mitigating issues such as subacute ruminal acidosis (SARA). Nonetheless, a more thorough discussion of the study’s originality in comparison to existing literature is lacking. The authors should clarify the novel contributions of this work relative to previous studies on Candida rugosa and other ruminal yeasts (e.g., were new functions discovered? Were more effective strains identified? Is there a novel practical application being proposed?).

Response 1:Thank you for your valuable feedback. We agree that a clearer explanation of the novelty of our work compared to existing literature is necessary, and we will revise the manuscript to emphasize this aspect more clearly.Regarding the originality of our study, it makes several unique contributions compared to previous research: Previous studies have isolated Candida  from the bovine gastrointestinal tract and confirmed its potential as a feed additive for cattle through specific physiological and cytological screening. However, these studies did not validate the practical effects of Candida rugosa in animal production through in vitro rumen fermentation experiments or in vivo animal trials. In our study, we isolated and screened Candida rugosa (NJ-5)as a probiotic candidate strain, which was then prepared into a yeast culture. Using in vitro rumen fermentation experiments, we systematically evaluated its effects on the fermentation characteristics of high-concentrate substrates. The results showed that Candida rugosa NJ-5 yeast culture effectively alleviated the pH decline in fermentation broth induced by high-concentrate substrates and improved rumen fermentation, suggesting its potential application in mitigating SARA induced by high-concentrate diets in animal production. We have added the details to the Discussion section (Page 12-13, Lines 364-376).

Comments 2: The selection of strain NJ-5 followed a rational process, including assessments of anaerobic growth, acid tolerance, and lactate utilization. However, the rationale for selecting NJ-5 over other strains is not sufficiently justified in terms of functionality. It is recommended to strengthen the explanation for why NJ-5 was chosen as the best candidate—not only in terms of growth performance, but also regarding fermentation parameters and microbiota modulation.

Response 2: We sincerely thank the reviewer for their valuable suggestion. In response, we have expanded the rationale for selecting Candida rugosa NJ-5 as the optimal candidate strain. While the strain’s robust anaerobic growth, acid resistance, and efficient lactate utilization were key factors in its selection, additional fermentation parameters were also considered. In our in vitro rumen fermentation trial, live-cell suspensions of Candida rugosa(NJ-5)demonstrated a significantly higher pH after 24 hours of fermentation compared to the control group, which indicates its effective ability to mitigate ruminal pH decline. Moreover, this strain significantly enhanced rumen fermentation patterns by increasing the proportion of acetate in the fermentation broth, further highlighting its functional superiority. Based on these comprehensive findings, we believe that Candida rugosa(NJ-5)is not only superior in terms of growth performance but also in optimizing rumen fermentation, thus justifying its selection as the best candidate for the intended application.We have added the details to the Discussion section (Page 13, Lines 390-400).

Comments 3:The data reveal interesting effects on pH, volatile fatty acids (VFAs), and the microbial community. However, the biological interpretation remains somewhat superficial. There is no discussion on why acetate levels increased while propionate decreased, nor on the biological significance of reductions in groups such as Cyanobacteria or increases in Desulfobacterota and Christensenellaceae_R-7. It is suggested to incorporate a more in-depth discussion on the functional implications for animal health and productivity.

Response 3: Thank you for your insightful comments. We appreciate your suggestion to enhance the biological interpretation of the data and discuss the functional implications more thoroughly.The supplementation with Candida rugosa(NJ-5)yeast culture led to an increase in the relative abundance of Desulfobacterota, a group of sulfate-reducing bacteria .These bacteria are capable of reducing sulfate to hydrogen sulfide, which might influence the rumen’s redox environment.The increase in Desulfobacterota may reflect a self-regulatory adaptation of the rumen microenvironment, which could be contributing to the observed changes in acetate and propionate production.Regarding the reductions in Cyanobacteria, we agree that their role in the rumen warrants more explanation. Cyanobacteria are typically present in trace amounts in ruminant microbiomes and are commonly found in soil and aquatic environments. Their functional role in the rumen is not fully understood, but their presence may be linked to limited oxygen diffusion or fermentation of polysaccharides under microaerobic conditions. The reduction in Cyanobacteria observed in our study could reflect changes in the oxygen availability or other environmental conditions within the rumen due to the supplementation with Candida rugosa yeast culture.Finally, the increase in Christensenellaceae_R-7, which is known to play a role in gut microbiota modulation, immune regulation, and metabolic homeostasis, could be particularly important for animal health and productivity. In ruminants, this group has been shown to enhance rumen development and improve nutrient digestion and absorption. The selective promotion of Christensenellaceae_R-7 observed in our study suggests that the yeast culture supplementation may have beneficial effects on host metabolism and overall productivity. We will include a more detailed discussion on the functional implications of this increase, emphasizing its potential impact on animal health and performance.We have added the details to the Discussion section (Page 15, Lines 471-487).

  We appreciate the reviewer's insightful comment regarding the contradictory reports on acetate and propionate modulation by yeast supplementation. As highlighted in the revised discussion (Page14 , Line443-457):Volatile fatty acids (VFAs) produced by rumen bacteria serve as a crucial energy source for ruminants. Adding yeast to the diet may change the types of bacteria in the rumen, helping the bacteria that break down fiber, which in turn changes the kinds and amounts of certain volatile fatty acids (VFAs) made in the rumen. Studies demonstrate that yeast supplementation in dairy cows can significantly increase the concentrations of acetate, propionate, and total VFAs . Propionate, as an essential gluconeogenic precursor, significantly influences energy metabolism in ruminants. The modification of rumen fermentation processes induced by high-grain diets is a key contributor to SARA . Research demonstrates that high-grain feeding modifies rumen fermentation from an acetate-dominant pattern to one characterized by propionate and butyrate dominance. This study revealed that yeast culture significantly increased total VFA and acetate levels while markedly reducing valerate; however, no significant changes were seen in the concentrations of valerate, isovalerate, and isobutyrate. The data suggest that yeast culture modifies the fermentation pattern of high-concentrate substrates from propionate dominance to acetate dominance, thereby partially mitigating the effects of high-concentrate diets.  

Comments 4:The experiment was conducted solely in vitro. It would be advisable to discuss the limitations of the in vitro model, such as the absence of absorption dynamics and rumen-intestine passage, and the differences between free-living and epimural bacterial populations. Additionally, the authors should propose directions for future in vivo studies.

Response 4:We sincerely appreciate your insightful comments regarding the limitations of our in vitro rumen fermentation model. In the revised manuscript, we have added a detailed discussion on this point in the Discussion section (Page 14, Lines 442-450) as follows:In vitro rumen fermentation is a widely used, simplified technique that effectively mimics real rumen fermentation, offering advantages such as operational simplicity, ease of standardization, and high reproducibility. However, we recognize that this method has inherent limitations. It cannot fully replicate the complex rumen environment within a single vessel, such as the absence of absorption dynamics, the rumen-intestine passage, and the differences between free-living and epimural bacterial populations. Since the experimental findings are derived from in vitro conditions, they may not fully reflect the in vivo rumen fermentation processes in live animals.To address these gaps, we acknowledge the necessity for additional in vivo studies to evaluate the potential applications of Candida rugosa(NJ-5). Such studies will be crucial for assessing the strain’s performance within the actual rumen environment, including its impact on nutrient absorption, rumen microbiota dynamics, and overall animal health. We hope that these future studies will provide a more comprehensive understanding of the strain's functionality.We have added the details to the Discussion section (Page 15, Lines 500-508).

Comments 5:Finally, in the conclusion section, it is recommended to reinforce the closing statements with practical perspectives and suggestions for future applications.

Response 5:We sincerely appreciate your valuable comments on this study. We fully agree with the need to reinforce the closing statements with practical perspectives and suggestions for future applications. In response, we have made the following key additions in the revised manuscript (Page 14, Lines 450-453):Therefore, rumen native yeast Candida rugosa (NJ-5) has significant potential in mitigating SARA induced by high-concentrate diets, providing both an excellent microbial strain and a theoretical basis for developing probiotics to prevent and treat SARA in ruminants. Given these promising findings, future research could explore the application of Candida rugosa (NJ-5) in in vivo studies to further assess its efficacy in practical livestock settings. Additionally, it may be beneficial to investigate the strain’s long-term impact on rumen health and animal performance, potentially contributing to the development of novel probiotics for sustainable livestock farming.We have added the details to the Discussion section (Page 15, Lines 518-520).

Reviewer 2 Report

Comments and Suggestions for Authors

The authors present a study in which 59 yeast strains were isolated from goat rumen fluid, screened for anaerobic growth, acid tolerance, and lactate utilization, and subjected to in vitro fermentation assays. Candida rugosa (NJ-5) was selected for dose–response evaluation of a yeast culture product (0–5% DM basis) on rumen fermentation parameters and microbial community composition. The study claims that NJ-5 yeast culture at 5% mitigates pH decline, increases total VFAs and acetate, and selectively enriches fibrolytic and beneficial bacterial taxa in vitro, supporting its probiotic potential.

The topic is timely and relevant: exploring autochthonous yeast strains for rumen health under high–concentrate feeding addresses a clear gap, given the limitations of S. cerevisiae in ruminants. The isolation-to-in vitro screening pipeline is well conceived. However, the manuscript requires substantial improvement in clarity, methodological detail, data interpretation, and presentation before it can be considered for publication.

Major Comments

The Introduction should more clearly articulate why goat‐derived yeasts may outperform commercial S. cerevisiae. Cite specific comparative studies. Emphasize the novelty: only one previous report used C. rugosa for rumen fermentation (Sirisan et al., 2013), but none have developed a culture product or conducted dose–response in vitro.

 Methodological Details

Yeast Quantification: How was CFU/mL of the yeast culture determined on a dry‐matter basis? Provide plating or cell‐count methods.
Fermentation Substrate: The substrate simulates a 70:30 concentrate:roughage diet, but the actual DM composition (starch, fiber fractions) should be tabulated rather than described in text.

Data Presentation and Interpretation

Tables: Replace superscript letters with consistent formatting; include exact p-values.
Figures: Aggregate low-abundance taxa (<1%) in an “Other” category; highlight functionally relevant taxa.
Discussion: Discuss contradiction of acetate increase vs. propionate in literature; speculate on yeast metabolites.

Ethical and Compliance Statements

Provide consistent protocol number and citation for animal ethics approval.

Specific Comments

- Line 15–17: Remove redundant phrasing in the title; refine for brevity.

- Line 24: Change “HYC groups can mitigate” to “HYC significantly mitigated”.

- Line 61: Cite specific in vitro protocols (e.g., Menke et al., 1979).

- Line 142: Provide OD600 values for each strain in a supplementary table.

- Line 169: Clarify whether replicates are biological or technical.

- Line 229: Numerically describe strain selection (e.g., “37/59 strains grew anaerobically”).

- Line 339: Condense repetitive Results narrative in Discussion.

Author Response

Comments 1: The Introduction should more clearly articulate why goat‐derived yeasts may outperform commercial S. cerevisiae. Cite specific comparative studies. Emphasize the novelty: only one previous report used C. rugosa for rumen fermentation (Sirisan et al., 2013), but none have developed a culture product or conducted dose–response in vitro.

Response 1: We appreciate this constructive suggestion. The introduction has been revised to: Prior research indicated that supplementation with active dry yeast failed to restore the original microbiota composition that existed before the commencement of SARA in dairy calves, implying restricted functional effectiveness of this commercial yeast strain. This evidence underscores that the efficacy of probiotics is contingent upon their adaptation to certain ecological niches. Candida has been recognized as a natural species in bovine gastrointestinal tracts, with its probiotic potential experimentally confirmed. As of now, the sole study by Sirisan et al. has shown the application of Candida rugosa in rumen fermentation research. Nevertheless, no subsequent research has produced culture products from Candida rugosa or conducted in vitro dose-response trials.(Page2, Lines 74-83).

Comments 2: Yeast Quantification: How was CFU/mL of the yeast culture determined on a dry‐matter basis? Provide plating or cell‐count methods.

Response 2: We appreciate the reviewer's inquiry regarding yeast quantification. We acknowledge an inaccuracy in the original description of yeast quantification methods in Line 166. To clarify: The yeast counting in our study was performed directly from yeast culture broth, not on a dry-matter basis as previously stated. All CFU/mL determinations were conducted using standard plate counting techniques. Detailed Methodology: 1. Sample preparation: Fresh yeast culture broth was used without drying. 2. Serial dilution: 10-fold dilutions in sterile PBS buffer. 3. Plating: 100 μL aliquots were spread on YPD agar plates. 4. Incubation: Incubation was for 48 hours at 30°C. 5. Calculation: Colonies counted (in the 30-300 range) and converted to CFU/mL of the original broth.

Comments 3:Fermentation Substrate: The substrate simulates a 70:30 concentrate:roughage diet, but the actual DM composition (starch, fiber fractions) should be tabulated rather than described in text.

Response 3:We appreciate the reviewer's suggestion.The textual description was initially provided for brevity, but we agree that a table would improve clarity.The composition and nutritional levels of the fermentation substrate are shown in supplementary table S1 in the revised manuscript(Page 16, Lines 523).

Comments 4:Tables: Replace superscript letters with consistent formatting; include exact p-values.

Response 4:Thank you for your valuable suggestion. We have revised the superscript letters in the tables to ensure consistent formatting. In addition, we have added the exact p-values below the tables to provide more precise information. These revisions are indicated on (Page8-10 , Line 276/289/298) of the revised manuscript.

Comments 5:Figures: Aggregate low-abundance taxa (<1%) in an “Other” category; highlight functionally relevant taxa.

Response 5:We appreciate this constructive suggestion. Currently, the figures and tables present the top 20 phyla and genus with relative abundances greater than 1%.

Comments 6:Discussion: Discuss contradiction of acetate increase vs. propionate in literature; speculate on yeast metabolites.

Response 6:We appreciate the reviewer's insightful comment regarding the contradictory reports on acetate and propionate modulation by yeast supplementation. As highlighted in the revised discussion (Page14 , Line443-457):Volatile fatty acids (VFAs) produced by rumen bacteria serve as a crucial energy source for ruminants. Adding yeast to the diet may change the types of bacteria in the rumen, helping the bacteria that break down fiber, which in turn changes the kinds and amounts of certain volatile fatty acids (VFAs) made in the rumen. Studies demonstrate that yeast supplementation in dairy cows can significantly increase the concentrations of acetate, propionate, and total VFAs . Propionate, as an essential gluconeogenic precursor, significantly influences energy metabolism in ruminants. The modification of rumen fermentation processes induced by high-grain diets is a key contributor to SARA . Research demonstrates that high-grain feeding modifies rumen fermentation from an acetate-dominant pattern to one characterized by propionate and butyrate dominance. This study revealed that yeast culture significantly increased total VFA and acetate levels while markedly reducing valerate; however, no significant changes were seen in the concentrations of valerate, isovalerate, and isobutyrate. The data suggest that yeast culture modifies the fermentation pattern of high-concentrate substrates from propionate dominance to acetate dominance, thereby partially mitigating the effects of high-concentrate diets.

Comments 7:Provide consistent protocol number and citation for animal ethics approval.

Response 7: We sincerely apologize for this oversight.We have standardized the ethics statement with the approval number: IACUC Approval No. NJAU.20221101207 from the Animal Ethics Committee of Nanjing Agricultural University, which has been added to the Methods (Page 2, Line 85) and reiterated in the "Ethics Declaration" section of the cover letter (Page 16, Line 534) .

Comments 8:Line 15–17: Remove redundant phrasing in the title; refine for brevity.

Response 8:Thank you for the suggestion. I have revised the title to remove the redundant phrasing and made it more concise.The title has been revised from:"Finally, Candida rugosa (NJ-5) with good probiotic characteristic was chosen to investigate its effect on ruminal fermentation in vitro. Batch culture technique was used to explore the effect of Candida rugosa (NJ-5) yeast culture on rumen fermentation parameters. "to:"The probiotic yeast Candida rugosa (NJ-5) was selected for in vitro batch culture studies on rumen fermentation." (Page 1,line 15-16).

Comments 9:Line 24: Change “HYC groups can mitigate” to “HYC significantly mitigated”.

Response 9: Thank you for the suggestion. I have changed 'HYC groups can mitigate' to 'HYC significantly mitigated' to more accurately reflect the findings of the study. (Page 1,line 26).

Comments 10:Line 61: Cite specific in vitro protocols (e.g., Menke et al., 1979).

Response 10:Thank you for the suggestion.As suggested by the reviewer, we have included the additional references( Page 2,line 64).

Comments 11:Line 142: Provide OD600 values for each strain in a supplementary table.

Response 11:Thank you for this suggestion. The OD600 values for all strains have been added as Supplementary Table S2(Page 16, Lines 523). .

Comments 12:Line 169: Clarify whether replicates are biological or technical.

Response 12:Thank you for the suggestion.As suggested by the reviewer, we have corrected the “with three replicates per group” to “ with three biological replicates per group” (Page 4,line 168-169).

Comments 13:Line 229: Numerically describe strain selection (e.g., “37/59 strains grew anaerobically”).

Response 13:Thank you for the suggestion.The anaerobic screening was based on evaluating yeast strains' ability to grow under anaerobic conditions, which could not be quantitatively described. The screening procedure was conducted as follows: All 59 isolated yeast strains were streaked onto YPD agar plates, which were then placed in anaerobic bags and incubated at a constant temperature for 48 hours. After incubation, colony growth was assessed. Strains exhibiting growth were classified as anaerobic-tolerant, while those showing no growth were designated as anaerobic-sensitive.

Comments 14:Line 339: Condense repetitive Results narrative in Discussion.

Response 14:Thank you for your insightful comment. We have carefully revised the Discussion section to condense repetitive descriptions of the Results. Redundant narrative has been removed or streamlined to enhance clarity and avoid duplication.

Round 2

Reviewer 2 Report

Comments and Suggestions for Authors

The manuscript has been corrected appropriately.